


# On the nonlinear and Solar-forced nature of the Chandler wobble in the Earth's pole motion

Dmitry M. Sonechkin

Shirshov Oceanology Institute, RAS, Moscow, Russia

5 *dsonech*@ocean.ru

**Abstract.** About 250 years ago L. Euler has derived a system of three quadratic-nonlinear differential equations to depict the rotation of the Earth as a rigid body. Neglecting a small distinction between the equatorial inertia moments, he reduced this system to much simpler linear one, and concluded that the Earth's pole must experience a harmonic oscillation of the 304-day period. Astronomers could not find this oscillation, but instead, S.C Chandler 10 has found two powerful wobbles with the 12- and ~14-month periods in reality. Adhering to the Euler's linearization, astronomers can not explain the nature of the later wobble up to now. I indicate that the neglect by the above small distinction ("a small parameter" of the Euler's primary nonlinear equations) is not admissible because the effect of this parameter is singular. Analysing the primary equations by an asymptotic technique, I demonstrate that the Chandler wobble tones are formed from combinational harmonics of the Euler's 304-day oscillation, long- 15 term Luni-Solar tides as well as the 22-year cycle of the heliomagnetic activity. Correlating simultaneous variations of the wobble and a solar activity index, I corroborate that the Chandler wobble is really affected by the Sun.

## Introduction

About 250 years ago, a member of the Russian Academy of Sciences L. Euler has derived his famous system of three quadratic nonlinear differential equations

$$20 \quad A\dot{\omega}_1 = (B-C)\omega_2\omega_3, \ B\dot{\omega}_2 = (C-A)\omega_1\omega_3, \ C\dot{\omega}_3 = (A-B)\omega_1\omega_2 \tag{1}$$

to depict the rotation of the Earth as a rigid body. Neglecting a small distinction between the principal inertia moments $A$ and $B$, he reduced Eq. (1) to a system of two linear equations

$$\ddot{\omega}_i = -\Omega_{rig}^2 \omega_i, \ i=1,2 \tag{2}$$

the general solution of which consists of a harmonic oscillation for the polar motion components

$$25 \quad \omega_i(t) = R\cos\left(\Omega_{rig}t + \Phi_i\right), \ i=1,2 \ . \tag{3}$$

The eigen frequency $\Omega_{rig} = \frac{C-A}{A}\Omega$ corresponds to the 304-day period of a hypothetic polar motion wobble in this solution.

However, astronomers could not found the 304-day period in the Earth's pole real motion. Instead, at the very end of the XIX century, an amateur of astronomy from USA. Chandler, 1891a has found two powerful wobbles with the 30 12- and approximately 14-month periods. Immediately, the nature of the 12-month wobble was explained by astronomers as a result of seasonal redistributions of air and water masses in the Earth's atmosphere and oceans. The 14-month wobble was criticized as unreliable firstly. Only after some re-examinations Newcomb, 1892 admitted that Chandler was right, and proposed to explain the discrepancy between the above Euler's solution and the really observed 14-month wobble as due to an elasticity of the Earth body. S.S. Hough, H. Poincare, F. Sludskii, V. 35 Volterra and some other famous scientists (see Lambeck, 1980; Munk and Macdonald, 1960; and Sidorenkov, 2009 for details) were the first who started to discover the Chandler wobble phenomenon. Their indications were that the main modes of the polar motion must be like a rigid rotation, and Volterra also has noted that the mantle elasticity must decrease but not increase the wobble period.



In spite of this, the explanation proposed by Newcomb is generally accepted up to now (e.g., see Gross, 2015). According to this explanation, the Chandler wobble is a phenomenon that may be treated in the frame of the Euler's linearization Eq. (2) if a dressed frequency $\Omega_{ela} = \frac{C-A-\Re}{A+\Re}\Omega$ is used instead of the bare Euler's frequency $\Omega_{rig}$ to take into consideration the effects of the mantle elasticity and differential rotations of the inner Earth's

cores ($\Re$ is a function of the so-called Love numbers measuring these effects). But, because some dissipative processes associated with the wobble-induced deformations of the Earth body cause the Chandler wobble to freely decay, it is accepted to add a forcing term (called the excitation function $\psi_i$, $i=1,2$) to Eq. (2):

$$\ddot{\omega}_i = -\Omega_{ela}^2\omega_i + \psi_i \ , \ i=1,2 \ . \tag{4}$$

Searching any excitation function which contains within itself some periodicities resonant to the eigen frequency

$\Omega_{ela}$, many astronomers drew attention to correlations between the real polar motion and the Earth's angular momentum variations induced by air and water re-distributions in the Earth's atmosphere and oceans as these re-distributions are seen in models or in observational data. A number of such correlations were already demonstrated, but new ones continue to appear Eq. (6) evidencing dissatisfaction of the presently working scientists with the correlations demonstrated earlier.

**1 Result**

The above linearization approach seems to be unsatisfactory because it is very difficult to comply with the dressing $\Omega_{rig}$ to $\Omega_{ela}$ without some artificial fitting of the Love numbers. Besides, no wobble is possible to observe at super- and sub-harmonic tones of $\Omega_{ela}$ in the frame of the linear system Eq. (4) although such tones undoubtedly exist in the real Earth's pole motion. In particular, the semi-Chandlerian period, the ~6-7-year long beat of the

annual and Chandlerian periods, and even more complex tones are registered in the real polar motion spectra (Carter, 1981; Wang, 2002; Hoepfner, 2004; Guo et al., 2005; Ivashchenko et al., 2006).

Thus, all these tones must be represented in the time-variable excitation function itself. However, there are two points which must be taken into consideration.

Firstly, let's suppose that the excitation function contains some necessary periodicities within itself, and a

combinational harmonic of these periodicities can resonate with the main Chandlerian periodicity (of the 433 – 435-day period) in principle. For example, Sidorenkov, 2009 has indicated such necessary periodicities. However, it is well-known that any linear dynamical system responds to the external periodicities separately. It means that the power spectrum of any linear dynamical system will reveal spectral density peaks at periods of each external periodicity, but no peak can be seen at the period of the combinational harmonic.

Secondly, the atmospheric and oceanic variations are known to be very broad-band in their character, and all published declarations about the existence of some proper periodicities in variations inherent to atmosphere and oceans are under great doubt. Besides, correlation is not a cause-effect relationship between quantities compared in principle, and so the afore-mentioned correlations (very subtle in fact) can evidence responses of the atmosphere and oceans to the polar motion wobble by the same right (Bryson and Starr, 1977; Egger and Hoinka, 1999). Then, it is

known (see: Plag et al., 2004) and many other publications) that the Chandler wobble period and amplitude are time-



dependent and might be positively correlated with each other as it is inherent to nonlinear dynamical systems, but usually absent in linear ones. At last, the air and water masses that can be involved into the re-distribution processes perhaps are less than the equatorial bulge mass excluded by the Euler's linearization, and so it is impossible to ignore a torque induced by this bulge in Eq. (4).

Thus, from the physical point of view it seems to be more appropriate to treat the polar motion as a nonlinear phenomenon (Sonechkin, 2001). For example, as it has been done in (Chen et al., 2009, Folgueira and Souchay, 2005), the general solution of the primary nonlinear equations Eq. (1) may be represented via Jacobi's elliptic functions. Unfortunately, this approach is applicable only to the free oscillations in the systems like Eq. (1), and the respective solution is not known for the forced polar motion. Therefore, it is preferable to transform the primary

Euler's equations Eq. (1) to an equivalent form with explicit representation of their "small parameter"

$$\varepsilon = (B - A)/C:$$

$$
\begin{aligned}
A\dot{\omega}_1 &= (A + \varepsilon C - C)\omega_2\omega_3, \\
(A + \varepsilon C)\dot{\omega}_2 &= (C - A)\omega_1\omega_3 \\
\dot{\omega}_3 &= -\varepsilon\omega_1\omega_2.B
\end{aligned}
\qquad (5)
$$

and then search an asymptotic expansion of the general solution of these new equations $\omega_i(t) = \varepsilon^0\omega_{i0}(t) + \varepsilon\omega_{i1}(t) + \dots$. It is clear that, neglecting in Eq. (5) all terms of the order $\varepsilon$, one can

obtain a linear system

$$
\begin{aligned}
A\dot{\omega}_1 &= (A - C)\omega_2\omega_3, \\
A\dot{\omega}_2 &= (C - A)\omega_1\omega_3 \\
\omega_3 &= const
\end{aligned}
\qquad (5a)
$$

It is easy to see that Eq. (5a) is another form of Eq. (2). Therefore, the zero-order approximation of the asymptotic expansion of the general solution of Eq. (5) consists of some harmonic oscillations identical to Eq. (3). Substituting this zero-order solution (marked $\omega_{i0}; i = 1,2,3$) into Eq. (5), and taking into consideration only the terms of the

order $\varepsilon$, one can obtain the following linear system

$$
\begin{aligned}
\ddot{\omega}_{11} &= -\left(\frac{C-A}{A}\right)^2\Omega^2\omega_{11} + \left\{2\frac{C-A}{A}R^3 + \frac{(C-A)C^2}{A^3}\Omega^2 R\right\} \\
&\cos\left(\frac{C-A}{A}\Omega t\right) - 2\frac{(C-A)}{A}R^3\cos\left(3\frac{C-A}{A}\Omega t\right), \\
\ddot{\omega}_{21} &= -\left(\frac{C-A}{A}\right)^2\Omega^2\omega_{21} + \left\{2\frac{C-A}{A}R^3 + \frac{(C-A)C^2}{A^3}\Omega^2 R\right\} \\
&\sin\left(\frac{C-A}{A}\Omega t\right) - 2\frac{C-A}{A}R^3\sin\left(3\frac{C-A}{A}\Omega t\right)
\end{aligned}
\qquad (6)
$$

The first terms in the right hand sites of Eq. (6) depict free oscillations at the frequency $\Omega_{rig}$, but the second and the third terms have a character of external forces. Omitting intermediate transformations because these are very





simple and cumbersome (the respective technique is stated in numerous textbooks like (Nayfeh, 1981), one makes

sure that the general solution of Eq. (6)

$$\omega_{i1}(t) = \alpha_{i1}\cos(\Omega_{Eu}t) + \beta_{i1}\cos(3\Omega_{Eu}t) + \mathbf{t}\boldsymbol{\gamma}_{i1}\cos(\Omega_{Eu}\mathbf{t}), \, i = 1, \, 2 \qquad (7)$$

includes a term (indicated in bold) with the time $\mathbf{t}$ as a co-factor. This term depicts a secular (infinitely growing)

5        contribution of the "small parameter" into the solution of the primary Euler's equations Eq. (1).

To exclude this term, a dependence of the oscillation frequency from the oscillation amplitude must be taken into

account. Such exclusion decreases the eigen frequency of the system Eq. (1) in comparison with the eigen frequency

of the linearized Euler's equations Eq. (2). However, this decrease is too small (of the order $\mathcal{E}$ ) to reach the periods

inherent to the real Chandler wobble (the main power spectrum peaks at ~435 and ~428 days, and more subtle peaks

near the periods of 400 and 450 days, see Eq. (7) and many other publications). Some effects of the long-periodic

beats of the Luni-Solar tides

$$G_i(t) = g_{ik}\cos(N_{ik}t)(1 + g_{im}\cos(N_{im}t)) \times \ldots, \, i = 1, \, 2 \qquad (8)$$

may be taken into account to reproduce properly of these real periods. Certainly, it is necessary to assume the

Earth's body elasticity so that these tide beats can affect the Earth's pole motion. But, taking in mind unreliability of

the estimations of the Love numbers being in our hands, one can suppose that the change $\Omega_{rig}$ for $\Omega_{ela}$ may

still be neglected in our first-order approximation Eq. (6). Besides, one can see from Eq. (6) that the action of Eq. (8)

on the Earth's pole motion is multiplicative in its character

$$G_i(t)\cos(\Omega_{rig}t), \, i = 1, \, 2 \, , \qquad (9)$$

and by this reason, some tones can be excited as sums and differences of $\Omega_{rig}$ and the frequencies inherent to the

Luni-Solar tides themselves. In particular, the difference between $\Omega_{rig}$ and the frequency of the semi-annual Solar

tide can produce the tone

$$\frac{1}{183} - \frac{1}{304} = \frac{1}{457} \, day^{-1} \, . \qquad (10)$$

This tone coincides well with one of the afore-mentioned subtle peaks of the real polar motion power spectrum.

Adding this tone by the frequency of the 8.85-year (3232-day) eastward precession of the Lunar node admits to

reproduce the tone of another subtle peak in the real power spectrum

$$\frac{1}{457} + \frac{1}{3232} = \frac{1}{400} \, day^{-1} \, . \qquad (11)$$

Adding Eq. (10) by the 18.66-year (6816-day) Lunar tide beat admits to reproduce the tone

$$\frac{1}{457} + \frac{1}{6816} = \frac{1}{428} \, day^{-1} \qquad (12)$$

that has been still indicated in the pioneering papers of Chandler, 1891b. Now this peak is estimated to be of the

second power in the real polar motion spectrum (Guo et al., 2005).

It has been, at first quite unexpectedly for the author of this paper, that the frequency of the most powerful

(according to the present-day estimations) power energy peak in the real polar motion spectrum (period of ~435





days) is reproduced quite well by a sum of Eq. (10) and the frequency of the Hale's 22-year (8036-day) cycle of the heliomagnetic activity

$$\frac{1}{457}+\frac{1}{8036}=\frac{1}{433}\ day^{-1}.$$

(13)

Unfortunately, although the afore-indicated periods are found in the Chandler wobble observations, it is difficult to compute these amplitudes based on the equations Eq. (6) and Eq. (8) solely.

Therefore, it must be a subject of the followed investigation.

An explanation of the heliomagnetic cycle effect may be expounded in the following way. Since the Earth is a magnet, the axis of this magnet must try to dispose along the Solar-induced magnetic field contours. However, the Earth's magnetic axis does not coincide with the Earth rotation axis. Its intersection with the Earth's surface was near 86.5 N, 72.62 W (near the Ellesmere island (Canada) at 2017, and moving to the direction of the Taimir peninsula (Siberia) with an acceleration. Thus, the Earth's magnet axis incessantly deviates from the above contours. Therefore, a torque must exist affecting the Earth's rotation. This torque itself is time-variable because of the Hale's heliomagnetic cycle (the Schwabe's doubled cycle of the Sun-spots). This cycle is not perfectly periodic. Its length varied from 15 up to 30 years over the time of its observations. Its amplitude also varied with a period of several decades. Moreover, the Solar-induced magnetic field shows phase catastrophes from time to time. In particular, such a catastrophe has been observed after 1923 (Polygiannakis et al., 1996; Duhau and Chen, 2002), at the same time moment when a phase catastrophe was found in the Earth's polar motion (Plag et al., 2005; Guo et al., 2005; Guinot, 1972; Guinot, 1982). The next such a catastrophe has been observed at the beginning of the XXI century (Malkin and Miller, 2010). The magnetic field catastrophe was prominent by an essential decrease of the yearly Sun spot number, and by shortening the Schwabe cycle length as well.

For the first time, an evidence of the heliomagnetic cycle effects on the Earth's pole motion has been given by Markovitz, 1960. Later, some other scientists confirm this evidence reality, and indicated that the heliomagnetic effect is a sum of the direct effect of the magnetic field on the Earth rotation and its indirect effect through changes in the atmospheric general circulation (Gross and Vondrak, 1999; Wang, 2004; Chapanov et al., 2009).

One can add my own evidence of the effect. For this, let us introduce a special Solar-activity index

$$SI(n)=0.5\big(MASC(n)\times LESC(n)\big)$$

(14)

to catch correlations between the Solar magnetic activity and the Chandler wobble period and radius with $MASC(n)$ be the maximal annual sunspot number in the Schwabe cycle $n$ and $LESC(n)$ be the length of this cycle. Fig1 shows that such correlations really exist. They are rather feeble (the dozes of the Chandler wobble variability explained are 0.28 and 0.19 respectively for the wobble period and radius) if the Schwabe cycles $n=13\div23$ are taken into consideration. But, if the Schwabe cycles $n=13\div15$ observed before the phase catastrophe of 1923 are rejected from consideration, the explained doses increase up to 0.96 and 0.88. It means that an almost functional relation exists for the quantities compared.

Finally mention that the real polar motion reveals a secular wander of the North geographic pole along ~80W near which the North geomagnetic pole (dipole pole) is located (80N,72W). It is a supplementary proxy evidence of a certain role of the magnetic field in the Earth's polar motion.

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

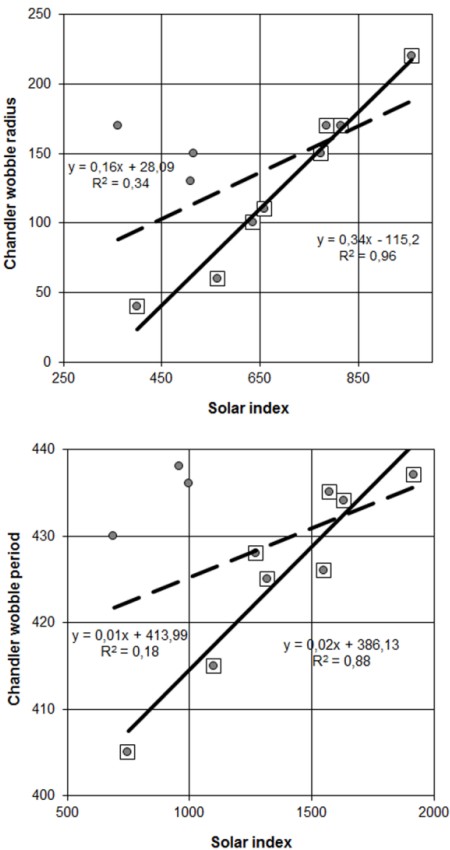

**Figure 1: Correlations between the Chandler wobble radius (top) and period (bottom) and the Solar index Eq. (14) for the later eleven (uninterrupted line) and eight (dotted line) Schwabe cycles.**