# Peer review of "On the nonlinear and Solar-forced nature of the Chandler wobble in the Earth's pole motion"

_Nonlinear Processes in Geophysics, 2019_

## Referee Comment (RC1) · Michael Efroimsky (Referee) · 7 May 2019

While the paper definitely contains an interesting and fresh idea, this idea had not been mathematically developed by the author to the level at which it is possible to judge on its viability. Accordingly, I am at this point unable to say whether the paper is publishable or not. Below, I offer some guidelines for the further work. If the author provides a development along these, I shall be able to consider a new version of his paper.

1. Referring to Newcomb (1892), the author suggests that the augmented Euler frequency is given by $\Omega (C-A-R)/(A+R)$. I am surprised to see the same addition '-R' both in the numerator and denominator. In the derivations, which I saw, the additions to the numerator and denominator turn out to be different.

For example, in equation (7.127) from the now-classical book https://doi.org/10.1017/CBO9781316136133 the corrections to the numerator and denominator do not coincide.

The theory of the Chandler wobble of the elastic homogeneous Earth is presented, in a remarkably simple language and with sufficient mathematical rigour, in the paper by Kubo: http://adsabs.harvard.edu/abs/1991CeMDA..50..165K From what I see there in eqn (3.11), the augmented Euler frequency of a homogeneous Earth looks as $\backslash$Omega (C - A - R) / A, with no correction in the denominator. This tells me that the elasticity enters the correction to the numerator, not the correction to the denominator (the latter correction comes from the layered structure solely). So these corrections are not obliged to coincide.

2. The author argues that a solution expressed through the Jacobi functions "is applicable only to the free oscillations in the systems" and that "the respective solution is not known for the forced polar motion". I see no ideological problem here. Some day, someone should start out from the solution in the Jacobi functions, add perturbations, and employ the method of variation of parameters. The resulting system will be cumbersome, but in principle this programme can be accomplished.

3. Based on the said argument, the author suggests to approximate the Euler system with eqns (5).

While I am open-minded enough to consider this approximation, I refuse to accept the author's motivation for it. The motivation came from the complaint that the Jacobi-function solution is inconvenient to be generalised to the perturbed case. However, in eqns (5), I see no perturbation. So the above motivation bears no relevance to the method.

Nonetheless, the method by itself is of interest and may be worth pursuing.

Sadly, the author stops at eqn (7). After that equation, the author states that the secular term may be excluded by taking into account the dependence of the frequency on amplitude. The idea is sound – but, alas, the author provides no mathematics in support of it.

The author points out that the difference between \Omega_{rig} and the frequency of the semi-annual solar tide can produce a tone present in the measured power spectrum. Combining this tone with the frequency of the 8.85-year (3232-day) eastward precession of the Lunar node, the author obtains another observable tone.

While these observations are intriguing, they can serve only as an addition to a clear mathematical development (a toy model, at least).

So, if the author wants to make this manuscript bona fide and publishable, he absolutely must introduce some perturbation, to model the lunar and solar torques and to demonstrate how to remove the secular terms. Without such a development, the paper will remain a collection of interesting hypotheses.

4. Combining the tone given by eqn (10) with the leading Chandler frequency, the author arrives at the Hale cycle of the heliomagnetic activity. He then suggests that the wonder of the magnetic pole must render a torque influencing the Earth rotation. This is a totally separate (and extremely complex) topic, and I would recommend the author to reserve it for a separate project – lest we get in this paper more hypotheses than proofs.

---

## Author Comment (AC1) · 15 May 2019

Item 1 I took the frequency (C-A-R)/(A+R) from Gross (2015) as an illustration only. It is not the aim of my paper to discuss what is the exact value of this FREQUENCY. Therefore, I just exclude this mention from the text.

Item 2 Certainly, the way indicated by the reviewer1 is possible to use. But I prefer to use another possible way via asymptotic expansion of the nonlinear Euler equation solution. I change the motivation of this indicated that, according to my knowledge nobody used the first way, at least in a geophysics yet. So the use the Jakoby functions is a way to uncertainty (no experience exists).

Item 3 Eq. (5) is added in the subsequent consideration in my text (external influences

on the system are considered (Eq. (8) and the next ones). In order to better motivate my choice I include into the text another representation of the nonlinear Euler system in the form of the cubic nonlinear oscillators (Dueffing's oscillators).. This representation is possible because the bare Euler system admits the existence of several first integrals. Under actions of external influences these integrals lose their time-invariability, and two first order ODEs must be added to depict the time evolution of the quantities. It is just "toy-model" asked by reviewer1. It already has been published by me, but in Russian only. This toy-model reveals that the external influences affect the Euler frequency. As a result, the Euler frequency varies in time in harmony with reality, but not constant as usually supposed.

Item 4 Of course, the topic of a possible influence of the solar activity on the polar motion is separate from the discussion of the Chandler wobble nonlinearity problem. It can be subject of another investigation. Unfortunately, I am afraid that I shall not be able to conduct such an investigation (because of my age). Therefore I write about this topic in this paper to stimulate younger researchers to develop this idea.

---

## Referee Comment (RC2) · Christian Bizouard (Referee) · 29 May 2019

At contemporaneous time scales (below 100 years) the non-linear terms commonly removed in Liouville equation cause negligible effects below 10 micro-arc-seconds, that cannot have any observable impact on the Chandler wobble. This conclusion can be easily reached by comparing numerical solutions of linear and non-linear Liouville equations written for an Earth composed of a quasi-elastic mantle and hydro-static oceans, and undergoing the hydro-atmospheric mass redistribution and luni-solar torque. Therefore I am very skeptical on the search for non-linear effects on polar motion over periods shorter than 100 years. The key point of the non-linear effect that the author wants to evidence is the perturbation of first order formalized by Eq. 7. But this perturbation remains undefined, not quantified in function of the input forcing. Whereas

the author try to explain the splitting of the Chandler band in a few harmonics through the influence of luni-solar tides, it does not prove anyhow to which extent this one can really excite the Chandler wobble, in this regard the Eq. (8) is totally esoteric. One of the most interesting aspect of the development given by the authors is the prediction of a free mode at frequency $3\Omega_{eu}$, where $\Omega_{eu}$ is, I guess, the Chandler frequency for a real Earth (many notations are not properly defined throughout this paper). This would imply a signal at 433/3   144 days in polar motion.

Furthermore the second part of the paper, investing possible link between Chandler wobble variability and solar activity, is quite decoupled from the first part of the paper. Moreover, the fact of eliminating three solar cycles for obtaining linear correlation with the amplitude of the period of the Chandler wobble is very questionable.

In conclusion I consider that the author has not proved anyhow that the Chandler wobbles results from "non-linear solar-forced" processes. Both mathematical developments and observational analysis are not convincing or not enough deepened.

---

## Author Comment (AC2) · 7 Jun 2019

Indeed, the small parameter of the nonlinear Euler's equations is quite small. Therefore, it isn't unexpected that solutions of these equations after their linearization look similar to solutions of the nonlinear equations if both kinds of the solutions are received numerically and if measures are taken for suppression of instabilities of the solutions of the nonlinear equations. However, it is strictly established fact in mathematics that a decrease in the order of the system of differential equations with a small parameter implies a qualitative change in the character of the system solutions. It is obviously evident that all solutions of the Euler system after its linearization are embedded into a two-dimensional plane. Such solutions can be either steady or pure periodic. Any complexity of these solutions can arise from an external forcing only. In contrast, the

solutions of the nonlinear Euler's system are embedded into a three-dimensional phase space. They can be complex with no external forcing in principle. Thus, I can not agree with the reviewer2 that the afore-mentioned solutions of the nonlinear and linear equations are similar with each other. Besides, one can mention that the Pole motion itself affects the atmospheric/ oceanic dynamics (the so-called Pole tides). Perhaps, a similar influence exists on the mantle dynamics. Therefore, some interactions between the Pole motion and the dynamics of other Earth's spheres must be taken into consideration when the Chandler wobble problem is analysed. The present-day models mentioned by the reviewer2 ignore this circumstance. So, they can not be properly reliable. In a corrected text of my paper I demonstrate that the nonlinear Euler's system can be represented by a sum of a Dueffing's cubic nonlinear oscillator and a regulator (Eqs. (9) and (10) in the corrected text). It is well known that solutions of the Dueffing oscillator are very complex. In particular, such form of the Euler's equation representation demonstrates that the momentary frequency of the oscillator vary in time. It well corresponds to the time-variable period of the Chandler wobble known from astronomical observations. Moreover, if an external forcing is taken into consideration the eigen frequency of the Dueffing oscillator is affected by this forcing multiplicatively. This fact represents a substantiation of the form of Eq. (8) (Eq. (11) in the corrected text). On this way I could obtain the period of 433 days in excellent agreement with observations when a solar activity effect was taken into consideration. It is why the latest part is included into the text of my paper even if the number of available heliomagnetic data is very limited, and the elimination of a part of these data is questionable really.

---

## Short Comment (SC1) · 21 Jun 2019

Sonechkin suggests that the wobble has some forced character to it. This should be obvious considering that only the annual cycle could create an annual response. So the 433 day cycle will be due to the lunar nodal fortnightly cycle (L) synchronized by the annual (Y) such that the wobble period is Y/(Y/L-26) = 432.7 days

please cite: Pukite, P., Coyne, D. & Challou, D. Mathematical Geoenergy: Discovery, Depletion, and Renewal. (Wiley, 2018). doi:10.1002/9781119434351

[Figure]

**Fig. 1.** Fit to lunar cycle

[Figure]

**Fig. 2.** Synchronization

---

## Author Comment (AC3) · 23 Jun 2019

I did not know the publication of Pukite et al. (2018). Of course, I include it into the reference list of the corrected text of my paper. However, I must say that there are numerous tidal periodicities which can affect the polar motion in principle. One can find several different combinations of these capable to produce time series more or less well reproducing the real pole motion. The combination found by Pukite et al. (2018) is one of such combination. In the primary text of my paper I already indicate another combination proposed by Sidorenkov (2009). Perhaps, even more such combinations can be proposed: At the same time, I agree that the choice of a concrete combination is debatable. However, I want to stress repeatedly the main message of my paper consists in the following. Any linear dynamical system responds to its external periodicities separately. It means that the power spectrum of the system reveals spectral density peaks at periods of each external periodicity, but no peak can be seen at the period of the combinational harmonic. The necessary and sufficient condition for the appearance a spectral peak at a combinational tone frequency consists of the dynamical system nonlinearity.

---

## Short Comment (SC2) · 24 Jun 2019

As is known from conventional tidal analysis used for predicting ocean sea-level heights, the forced responses must be (and are) numerically exact in terms of matching periodicities. Any of the lunar cycle combinations that do not match can be eliminated. My point is that there is only one lunar cycle possibility that works precisely for the Chandler wobble cycle, and since it is exact, it must be considered as the null hypothesis for any alternative model (such as a natural resonant response).

There is a practical analogy for this: consider an electrical circuit that exhibits a characteristic noise. If when that noise is measured it matches precisely the 60 Hz mains frequency, then that noise is most likely the result of inadequate filtering of the mains,

and other alternative models should be dismissed as low-likelihood possibilities.

So the original hypothesis of the Chandler wobble derived from Euler's original prediction of 305 days was actually a good characterization of the system's band-pass behavior, with the (exact) 365 day annual and (exact) 433 day lunar nodal period manifested as a forced response passed through the filter response window. The Q-factor of that window does not necessarily have to be high.

Overall, I am in agreement with Professor Sidorenkov's idea (which we have cited in the book) but only desire that the precision must be stressed, as that will provide a benchmark to compare against any alternative models of greater complexity. This seems to be a fundamental geophysics model that has been overlooked in the research and I want to see the strongest supporting argument backed with evidence presented in the paper. At some point this model should be as well accepted as the lunisolar model for ocean tides.

---

## Author Comment (AC4) · 25 Jun 2019

DMITRY SONECHKIN

dsonech@ocean.ru

All of these are right. But, extternal periodicities affect the systems additively in both of your examples. In contrast, external periodicities act multiplicatively in the case of the Pole motion. It can be seen if the fully nonlinear Euler's system is transformed into a system consisting of a cubic nonlinear oscillator (like the Dueffing oscillator) with a regulator. External periodicities affect the regulator, and only then the regulator changes the momentary frequency of the oscillator to be equal to the Chandlerian frequency. A preliminary consideration of this matter has been given in my old paper (2001) in Russian. Now, I add several sentences about this into the corrected text of my paper under discussion.